# Peristomal Skin Complications: Detailed Analysis of a Web-Based Survey and Predictive Risk Factors

**DOI:** 10.3390/healthcare11131823

**Published:** 2023-06-21

**Authors:** Eliana Guerra, Francesco Carlo Denti, Cristina Di Pasquale, Francesca Caroppo, Luisa Angileri, Margherita Cioni, Aurora Parodi, Anna Belloni Fortina, Silvia Ferrucci, Martina Burlando

**Affiliations:** 1Ambulatorio Riabilitazione Enterostomale, ASST Spedali Civili Brescia, 25123 Brescia, Italy; guerraeliana1971@gmail.com; 2Stoma Care Unit, San Raffaele Scientific Institute, 20132 Milan, Italy; denti.francescocarlo@hsr.it; 3Stomal Therapy Outpatient Service, European Institute of Oncology IRCCS, 20141 Milan, Italy; cristina.dipasquale@ieo.it; 4Unit of Dermatology, Department of Medicine DIMED, University of Padua, 35128 Padua, Italy; francesca.caroppo@unipd.it (F.C.); anna.bellonifortina@unipd.it (A.B.F.); 5Unit of Dermatology, Foundation IRCCS Ca’ Granda Ospedale Maggiore Policlinico, 20122 Milan, Italy; luisaangileri@hotmail.it (L.A.);; 6St Mary’s Hospital, Imperial College NHS Trust, London W2 1NY, UK; margherita.cioni@nhs.net; 7Di.S.Sal. Section of Dermatology, IRCCS San Martino Polyclinic Hospital, Largo Rosanna Benzi 10, 16100 Genoa, Italy; aurora.parodi@unige.it

**Keywords:** peristomal skin complications, stoma, risk factors, preventive measures, multidisciplinary approach

## Abstract

Patients with a stoma are at risk of developing peristomal skin complications (PSCs) that can negatively impact their quality of life. This study aims to identify potential risk factors for dermatitis, pruritis/xerosis, infections, and ulcerations among patients with a stoma and evaluate preventive measures. This cross-sectional study involved data regarding 232 Italian patients with a stoma. A questionnaire was used to collect patient characteristics, comorbidities, and stoma management data. The most frequent PSCs observed were dermatitis and pruritis/xerosis in approximately 60% of patients. Psoriasis was strongly correlated with dermatitis, while being overweight or obese increased the risk of pruritis/xerosis. Class 2 obesity and atopic dermatitis were associated with an increased risk of infections. Being underweight, completely nonautonomous, and having inflammatory bowel disease were associated with a higher risk of ulcerations, while radiotherapy was a strong risk factor for ulceration. Preventive measures such as using hydrocolloid barriers, TNT gauze cleansing, and low pH detergent were effective in preventing dermatitis. Appropriate stoma care and maintenance, including the use of protective film and careful monitoring of weight and comorbidities, are crucial in minimizing the risk of complications associated with a stoma.

## 1. Introduction

The creation of an abdominal stoma or ostomy is a frequently performed surgical procedure used to treat various benign and malignant pathologies, such as colorectal cancer, inflammatory bowel disease, diverticular disease, bowel injuries, ischemic colitis, and fecal incontinence [1]. In the United States, it is estimated that over 750,000 people live with an ostomy, and around 130,000 new ostomies are performed each year [2]. In Europe, there are approximately 100,000 patients with ostomies in Germany and 70,000 in Italy alone [3].

Stoma and peristomal complications are prevalent, ranging from 3% to 82%. These complications can be classified as early or late. Within the first 30 days following surgery, early complications may include stomal ischemia, mucocutaneous separation, retraction, and parastomal abscess [4]. Late complications, occurring after 30 days post-surgery, may include varices, prolapse, retraction, parastomal hernia, and stenosis [4]. Peristomal skin complications (PSCs) can also occur either early or late and are common after ostomy surgery [5]. The risk of PSCs depends on various factors, such as the underlying disease, the surgical procedure, and the patient’s characteristics [6]. All PSCs have a substantial negative impact on patients’ quality of life and pose a significant economic burden [2,3].

While surgical complications following stoma surgery have received significant attention, less emphasis has been placed on peristomal skin complications (PSCs) [4]. Many PSCs are preventable, and adequate training and active patient engagement can help limit their impact and severity [7,8]. However, PSCs are a significant challenge for a large number of patients with stomas [9], and they place a considerable burden on healthcare systems [10]. Unfortunately, one study found that while 45% of patients with an ostomy had a skin problem, only 43% were aware of it, and only 16% had sought treatment [11]. Additionally, little research has focused on identifying predictive risk factors for PSCs, which could be crucial in reducing their incidence through early identification, monitoring, and prevention [12].

The currently limited attention regarding predictive risk factors for PSCs can lead to an increased burden on healthcare systems and a negative impact on patients’ quality of life [7,8,12]. Herein, we present the results of a web-based survey of stoma nurses designed to collect the demographic and clinical characteristics of patients with an ostomy in Italy. Therefore, in addition to collecting extensive information on the presence and type of PSC and focusing on the predictive factors at baseline, the study aimed to develop a risk profile to aid in identifying individuals at higher risk of PSCs.

## 2. Materials and Methods

### 2.1. Design

The design of this study was a descriptive observational design that employed a web-based survey to collect data. The study followed the Checklist for Reporting of Survey Studies (CROSS) to ensure consistent reporting of survey data [13]. The lack of 4 months of follow-up was the only exclusion criterion.

### 2.2. Context, Ethics, and Procedure

The data were collected using a questionnaire compiled by 23 stoma nurses representing different geographical areas in Italy (i.e., convenience sampling). These nurses participated in a continuing medical education (CME) project of the Skin Health Academy (Italy) led by four dermatologists who received extensive training in preventing complications of peristomal skin and agreed to participate in this study. The survey utilized a methodology that ensured the anonymity of the patients [14,15]. The survey structure classified patient age and other demographics into discrete classes rather than continuous variables, following the principles of confidentiality and privacy [14,15]. This approach was carefully designed to strike a balance between the need for accurate information and the ethical imperative of protecting patient confidentiality. As a result, the survey did not require informed consent from the patients, even if the project had to be evaluated by the directions of the hospital where the involved nurses were employed. In addition, the Skin Health Academy board in Italy approved the project.

The data collection platform was active from 18 April 2022 to 7 October 2022. The nurses were requested to enter information on patients they had cared for during that time, including those with an ostomy created in the preceding 6–12 months and with at least 4 months of follow-up. In addition, all participating nurses administering the survey for gathering data about their patients had a standardized base of competence and had attended the same certifications in stoma care. Therefore, a nested structure in considering the data was considered unlikely, and the data were considered at an individual level without the need for nested models.

### 2.3. Measurements

The questionnaire contained 29 items that gained full consensus among authors to investigate meaningful aspects focused on peristomal skin disorders and covered patient characteristics and habits, previous or current therapies, type of ostomy, stoma care, follow-up, and the presence of a stoma therapist during treatment.

The baseline values nurses had to add to the survey were related to the time between the stoma creation and the following 14 days. Four types of PSCs were considered: dermatitis, pruritus/xerosis, infections, and ulcerations [16,17,18,19]. The baseline variables collected in this study aimed to provide a comprehensive overview of patient demographics and clinical characteristics using variables that were identified in a recent consensus [20]. These variables included age (categorized in years), sex (male, female), nutrition (low residue diet, diet for ileostomy, other), physical activity (never, almost never, sometimes, often), comorbidities (inflammatory bowel disease, IBD, diabetes, others), therapy at baseline (including chemotherapy and radiotherapy), type of stoma (colostomy, ileostomy, urostomy), and accessories for stoma care (such as remover, powder, and others). The stoma’s abdominal profile was also rated as flat/regular, or inward or outward shaped [21]. The plaques/skin barriers were defined according to international consensus criteria [22]. Additionally, the type of device used for ostomy care was noted, with options for one piece (where the adhesive wafer and pouch are combined into a single unit) and two pieces. Information on the material and shape of the plaque used for ostomy care was also collected. Habits in cleansing the stoma were documented, including the use of gauze or other methods, as well as the pH of the detergent used. The presentation of the stoma was also noted, with options for normal, everted, and other. BMI was also collected and classified into eight categories based on the BMI range, ranging from underweight to extreme obesity Class V [23].

### 2.4. Statistical Analysis

The statistical analyses were performed using IBM SPSS Statistics for Windows, Version 28.0 (Armonk, NY, USA: IBM Corp). Initially, a descriptive analysis was carried out, consisting of frequency distributions for categorical variables and measures of central tendency and variability for numeric variables. The first outcome, i.e., dermatitis/xerosis, was chosen as an anchor point to determine the sample size needed to apply the logistic regression model. For this outcome, we hypothesized a proportion of patients with outcomes equal to 60%, and we hypothesized a number of independent variables selected by the stepwise regression equal to 10 variables. Based on these hypotheses, the adequate sample size is equal to 250 patients. The valid sample size obtained after sampling is very close to this threshold, or 232 patients. Four separate forward stepwise logistic regression models were employed to identify significant predictors of dermatitis, pruritus/xerosis, infections, and ulcerations (outcomes). Independent variables were included in the models based on their *p*-values in relation to the outcome, and their selection was further supported by a literature search to verify the hypothetical clinical and biological association between the independent variables and outcomes. This combined approach represents a tradeoff between a data-driven approach and a theory-driven approach, allowing for a comprehensive selection of independent variables that considered both statistical significance and prior knowledge from the literature. A *p*-value threshold of 0.10 was chosen to select variables for inclusion in the final model, considering only those variables plausible as risk factors from a medical perspective [24]. The 0.10 *p*-value threshold was used only for selecting potential predictors. This approach is consistent with the need to select independent variables that are plausible risk factors for the outcomes of interest while avoiding unfitted models. By choosing a more lenient threshold while still considering only plausible risk factors, the models could include sufficient variables without compromising their fit. Notably, a significance level of 0.05 was employed while running the models. In each model, positive estimated coefficients (i.e., odds ratio [OR] > 1) were interpreted as risk factors for complications related to baseline characteristics, while negative estimated coefficients (i.e., odds ratio 0 > [OR] > 1) were considered protective factors. The odds ratios were reported with their corresponding 95% confidence intervals (95% CIs). The goodness of fit of each logistic regression model was evaluated using Nagelkerke’s pseudo-*R*^2^, comparing the correct classification rate between the naïve models and the estimated models and calculating the sensitivity and specificity of the estimated model. For analytics, missing data were managed with available case analysis.

## 3. Results

### 3.1. Patient Characteristics

A total of 23 stoma nurses participated in the survey, providing data on 263 patients (Figure 1). Each nurse contributed an average of 10 patients (range 7–12). After excluding 31 patients who had less than 4 months of follow-up, the final dataset included 232 patients.

Table 1 provides the demographic and clinical characteristics of the cohort.

The median age of patients was 65–70 years, and the mean post-intervention duration was 9.17 months (standard deviation = 3.79, range 4–30 months). The male-to-female ratio was 51.7% to 48.3%. Half of the patients had a normal BMI. Nearly 40% of patients had cardiovascular pathology at baseline, while 12.5% had diabetes. IBD, hematological pathology, tumor, psoriasis, atopic dermatitis, and allergic/irritant dermatitis were present in less than 10% of patients. More than 40% of patients received chemotherapy at baseline, and around 20% received radiotherapy or other therapies. Most patients experienced dermatitis (62.5%) and pruritis/xerosis (60.3%), while 10.8% had an infection and 35.3% had ulceration.

### 3.2. Logistic Models

Forward stepwise logistic regression analysis was then used to evaluate the association of individual patient characteristics on each of the four complications considered: dermatitis, pruritis/xerosis, infections, and ulcerations.

#### 3.2.1. Dermatitis

The regression analysis results are presented in Table 2, which outlines the variables significantly associated with dermatitis. The final regression model, which was obtained in step 14 and compared to the model with only the intercept (step 0), showed a marked improvement in the percentage of correct classification (from 62.3% to 79.1%) and a Nagelkerke’s pseudo-*R*^2^ of 44.1%. The specificity and sensitivity of the final model were 67.5% and 86.1%, respectively.

Utilizing a low pH detergent, gauze for cleansing, and a hydrocolloid plaque reduced the odds of dermatitis by 99% (OR = 0.013; 95% CI = 0.002–0.091; *p*-value < 0.001), 89% (OR = 0.109; 95% CI = 0.016–0.727; *p*-value = 0.021), and 89% (OR = 0.108; 95% CI = 0.041–0.282; *p*-value <0.001). Likely, the low residue diet was associated with a 78% decreased odds of dermatitis (OR = 0.222; 95% CI = 0.084–0.587; *p*-value = 0.002), followed by having a urostomy (OR = 0.251; 95% CI = 0.093–0.068; *p*-value = 0.007) and a regular abdominal stomal profile (OR = 0.314; 95% CI = 0.140–0.702; *p*-value = 0.005). The risk factors for dermatitis were psoriasis (OR = 33.324; 95% CI = 2.175–510.555; *p*-value = 0.012), having a one-piece device (OR = 3.763; 95% CI = 1.536–9.219; *p*-value = 0.004), and chemotherapy (OR = 2.237; 95% CI = 1.084–4.623; *p*-value = 0.029).

#### 3.2.2. Pruritis/Xerosis

The variables significantly associated with pruritis/xerosis are listed in Table 3. The final regression model obtained in step 9, compared to the model with only the intercept (step 0), showed an improvement in the percentage of correct classification (from 59.1% to 70%) and a Nagelkerke’s pseudo-*R*^2^ of 22.8%. The specificity and sensitivity of the final model were, respectively, 51.1% and 83.1%.

Sedentary lifestyles and age higher than 75 years reduced the odds of pruritis/xerosis by 76% (OR = 0.240; 95% CI = 0.073–0.792; *p*-value = 0.019) and 62% (OR = 0.383; 95% CI = 0.184–0.797; *p*-value = 0.010), respectively. Other risk factors for pruritis/xerosis were having a deep convex plaque (OR = 5.119; 95% CI = 1.060–3.963; *p*-value = 0.033), being a male (OR = 2.954; 95% CI = 1.584–5.507 *p*-value = 0.001, and being overweight or obese (OR = 2.341; 95% CI = 1.195–4.583; *p*-value = 0.013).

#### 3.2.3. Infections

Table 4 summarizes the variables associated with infections. The final regression model, obtained in step 7, showed an improvement in the percentage of correct classification (from 89.5% to 91.8%) and a Nagelkerke’s pseudo-*R*^2^ of 33.8%. The specificity and sensitivity of the final model were, respectively, 99.5% and 26.1%.

The use of a one-piece device was found to be protective against infections as it reduced by 83% the odds of reporting an infection (OR = 0.174; 95% CI = 0.036–0.840; *p*-value = 0.029). On the other hand, individuals with Class 2 obesity had a 30-fold increased risk of infections compared to those with normal BMI (OR = 30.313, 95% CI = 5.744–159.974; *p*-value = 0.001). The use of other medical therapies (OR = 4.473; 95% CI = 1.527–13.100; *p*-value = 0.006) and the presence of a hernia (OR = 11.818, 95% CI = 11.818; 95% CI = 2.257–61.883, *p*-value = 0.003), being completely dependent on a caregiver (OR = 2.989, 95% CI = 1.046–8.538; *p*-value = 0.041), and frequent physical activity (OR = 2.126; 95% CI = 1.084–4.170; *p*-value = 0.028) were all associated with an increased likelihood of infections.

#### 3.2.4. Ulcerations

Table 5 shows the variables associated with ulcerations. The final regression model obtained in step 10, compared to the model with only the intercept (step 0), showed an improvement in the percentage of correct classification (from 63.9% to 76.4%) and a Nagelkerke’s pseudo-*R*^2^ of 34.3%. The specificity and sensitivity of the final model were, respectively, 84.8% and 61.5%.

Protective factors for ulcerations included a normal abdominal stoma profile (OR = 0.391; 95% CI = 0.190–0.801, *p*-value = 0.009) and everted stoma (OR = 0.199; 95% CI = 0.058–0.684; *p*-value = 0.011) that reduced odds of ulcerations by 61% and 81%, respectively. In addition, the use of a protective film as an accessory (OR = 0.421; 95% CI = 0.207–0.854; *p*-value = 0.018) reduced the odds of ulcerations by 58%. On the other hand, being underweight (OR = 3.907; 95% CI = 1.419–10.758; *p*-value = 0.008), completely caregiver dependent (OR = 3.376; 95% CI = 1.645–6.928; *p*-value = 0.001), and the presence of IBD (OR = 4.161; 95% CI = 1.307–13.248; *p*-value = 0.016) were identified as significant risk factors for ulcerations. Allergic/irritant dermatitis (OR = 2.345; 95% CI = 0.842–6.531; *p*-value = 0.077), ileostomy (OR = 1.860; 95% CI = 0.906–3.819; *p* = 0.091), and radiotherapy (OR = 1.860, 95% CI = 1.055–5.436; *p*-value = 0.037) were also found to be associated with increased odds for ulcerations.

## 4. Discussion

This analysis provides detailed characteristics of patients with a stoma, including the most frequent skin complications observed at baseline, which were dermatitis and pruritis/xerosis in approximately 60% of patients. These findings are consistent with a previous study that reported peristomal irritant contact dermatitis as an early and late complication in 32% and 26% of patients, respectively [25]. Additionally, we identified potential risk factors for complications using the extensive patient data collected, focusing on dermatitis, pruritis/xerosis, infections, and ulcerations.

The presence of psoriasis was found to have a strong correlation with dermatitis. This is not surprising, as previous studies have shown that psoriasis and atopic dermatitis may coexist, have a longitudinal relationship, and share overlapping features [26,27]. Unlike a previous study [25], we did not find ileostomy to be a risk factor for dermatitis but rather found urostomy to be protective in our cohort. Our analysis also revealed that the use of hydrocolloid barriers, TNT gauze cleansing, and low pH detergent were all effective in preventing dermatitis. As per previous evidence, these findings demonstrate the importance of proper stoma cleansing and the use of hydrocolloid skin barriers, which have effectively reduced the incidence of dermatitis in patients with a stoma [21,28].

Regarding pruritis/xerosis, our analysis revealed that having a normal BMI was a protective factor, while being overweight or obese increased the risk of these complications. Interestingly, a sedentary lifestyle and age over 75 years were also found to be protective factors. Older age may be associated with less physical activity, which could explain this correlation. Reduced physical activity may lead to less leakage onto the skin, thus reducing the risk of pruritis/xerosis. In addition, our analysis found that males were at a higher risk for these complications. This association may reflect a slightly different distribution of some potential risk/protective factors between sexes, and in-depth research should further clarify these aspects. In addition, older individuals may have reduced sensitivity at the skin level, which could contribute to their reduced reporting of these symptoms [29].

Class 2 obesity was found to be strongly associated with an increased risk of infections, which is not surprising given that obesity is a well-known risk factor for infections [30]. Additionally, atopic dermatitis was also identified as a risk factor for infections, which is consistent with previous literature [31]. While peristomal hernia has been reported as a late complication of a stoma, it has not previously been associated with infections [6]. On the other hand, frequent physical activity and nonautonomy were found to be high-risk factors for infections. In contrast, the only protective factor identified was the use of a one-piece device, which is different from dermatitis, where a one-piece device was seen to place patients at a higher risk for that complication.

The analysis revealed that being underweight, completely nonautonomous, and the presence of IBD were all associated with a higher risk of ulcerations. Consistent with previous research, radiotherapy was also strongly associated with a greater risk for ulceration, as the majority of patients undergoing radiotherapy will experience skin injuries/ulceration [32]. In addition, the study found that both a normal and everted stoma profile were protective of ulcerations, as was the use of protective film and a normal abdominal profile. These findings suggest that appropriate stoma care and maintenance, including the use of protective film and careful monitoring of weight and comorbidities, are crucial in minimizing the risk of ulcerations in patients with a stoma.

The information on risk factors presented in this analysis and the related statistical analysis can be valuable for developing a tool to calculate individual risk. By entering a patient’s sex, BMI, and current comorbidities, the tool could estimate their potential risk of complications. This implication would be of significant clinical value in identifying the specific parameters that require close monitoring and maintenance to minimize the risk of complications associated with a stoma. It is important to note that the model developed would require validation and further refinement through additional studies. However, the potential benefits of such a tool in improving patient outcomes and reducing healthcare costs cannot be overstated.

This study sheds light on the prevention of PSCs, which thus far has received little attention. Few articles discuss prevention, including the SACS Evolution and the Ostomy Skin Tool 2.0, both of which are in the validation phase [32,33]. Although the present analysis was limited by the relatively small number of patients, the extensive data collection allowed for identifying potential risk factors associated with complications in patients with a stoma. Our findings emphasize the importance of outpatient follow-up by a specialized nurse who can advise on peristomal skin care using some of the risk factors identified in this analysis to drive patient care. Implementing tools that focus on PSC prevention could lead to improved patient quality of life, reduced clinical burden, and lower healthcare spending. Furthermore, the collaboration between nurses and a panel of dermatologists is innovative in the field of PSC prevention and highlights the importance of a multidisciplinary approach in the management of ostomy patients.

This study has several limitations that must be acknowledged. First, the convenience and non-probabilistic sampling methods used to recruit nurses may limit the generalizability of the findings to other populations. Second, the involved nurses added the data collected on patients’ characteristics and outcomes in the survey form; therefore, the investigators did not directly retrieve data from source documents such as clinical records, which could have led to potential inaccuracies. Additionally, the cross-sectional nature of the imputed data did not allow for the evaluation of the longitudinal trajectory of the investigated associations. Another limitation is that due to the small number of events for each specific type of dermatitis, it was not possible to conduct subgroup analyses to differentiate between the different types of dermatitis. In order to maintain a manageable number of independent variables in the logistic regression model, given the limited number of observed statistical units, and to mitigate issues related to multicollinearity, we adopted the stepwise technique to select only those risk factors with statistically significant impact on the dependent variable. However, it is important to note that this approach may result in excluding certain factors that could be relevant risk factors in the population. Future studies could consider collecting data on a larger sample of patients, including all potential risk factors, regardless of their statistical significance, to address this limitation and further explore potential risk factors. Lastly, it is important to acknowledge that the data used in this study were primarily entered and managed by nurses involved in caring for the patients. While the likelihood of a hypothetical nurse-outcome relationship was considered highly limited when the study was designed, it is necessary to recognize that the potential impact of the individual nurse who cared for the patient and entered the data cannot be completely disregarded. However, considering the standardized base of competence of the involved nurses as a framework, the collected data were considered at an individual level, which was deemed appropriate for the nature of this study. This approach allows for future designs to incorporate more robust methodologies. Although we acknowledge this limitation, it is important to note that as an observational study, our analysis focused on the relationship between the identified risk factors and the outcomes of interest, without explicitly examining the impact of individual nurses as a separate risk factor. For example, future studies may expand the focus of the investigation that in the current study was primarily on identifying predictive factors at baseline for PSCs. Although we also acknowledge the potential impact of factors such as the length of follow-up and the individual nurse’s involvement in patient care, our study design did not explicitly address these aspects. For all these reasons, future studies incorporating longitudinal data and explicitly considering the role of individual nurses in patient care would provide a more comprehensive understanding of the risk factors and outcomes associated with PSCs by expanding the scope of the current research.

Despite these limitations, this study provides valuable insights into the risk factors and preventive measures associated with complications in patients with a stoma. The insights gained from this study have the potential to inform clinical practice and guide healthcare professionals in effectively managing and improving the outcomes of patients with a stoma. The study aimed to develop a risk profile that can assist healthcare professionals in identifying individuals at a heightened risk of complications by emphasizing the baseline characteristics of patients. This approach offers a practical tool to proactively address potential complications and implement preventive measures, ultimately leading to improved patient outcomes, even acknowledging that additional meaningful elements could improve this approach. Furthermore, this work sets the stage for further exploration and expansion of research in this area, providing a foundation for future investigations to incorporate longitudinal data and explore the dynamic nature of complications in patients with a stoma. The insights gained from this study pave the way for a broader scope of research, enabling the development of more comprehensive strategies to optimize patient care and enhance long-term outcomes.

## 5. Conclusions

This study presented a detailed characterization of patients with a stoma, including the most frequent PSCs and potential risk factors associated with dermatitis, pruritis/xerosis, infections, and ulcerations. The results suggest that appropriate stoma care and maintenance, including the use of protective film and careful monitoring of weight and comorbidities, are crucial in minimizing the risk of complications associated with a stoma. The potential benefits of developing a tool to calculate individual risk based on a patient’s sex, BMI, and current comorbidities could improve patient outcomes and reduce healthcare costs. Implementation of tools that focus on prevention could lead to improved patient quality of life, reduced clinical burden, and lower healthcare spending. Future research should be designed to increase the external validity of the emerging findings, such as involving probabilistic sampling procedures and exploring the longitudinal trajectory of the investigated associations over time. Ultimately, a multidisciplinary approach involving specialized nurses and dermatologists is crucial in managing and preventing complications associated with a stoma.

## Figures and Tables

**Figure 1 healthcare-11-01823-f001:**
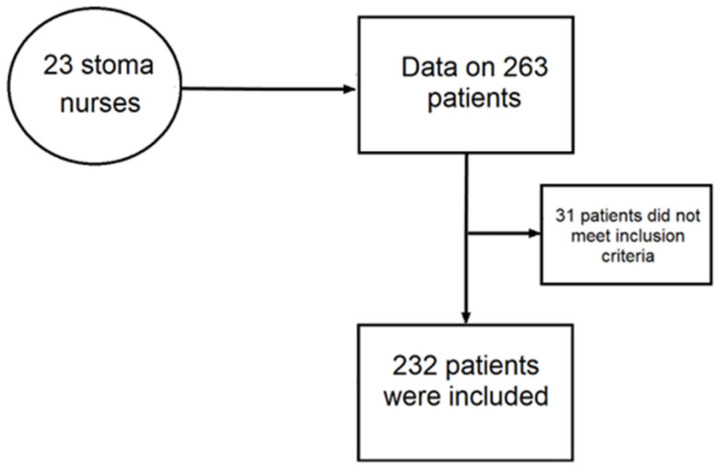
Patient selection flow for the final analysis.

**Table 1 healthcare-11-01823-t001:** Demographic and clinical characteristics of patients at baseline.

Main Characteristics	N	%
Age		
Median (years)	65–70 years	-
Time after intervention (months) [mean, standard deviation, SD]	9.17	3.79
Sex		
Female	112	48.3
Male	120	51.7
BMI		
Severely underweight	5	2.2
Moderately underweight	35	15.1
Normal	116	50.0
Overweight	49	21.1
Class 1 obesity	18	7.8
Class 2 obesity	9	3.9
Nutrition		
Low residue diet	38	16.4
Diet for ileostomy	84	36.2
Not applicable	110	47.4
Comorbidities		
IBD	21	9.1
Cardiovascular pathology	91	39.2
Hematological pathology	12	5.2
Diabetes	29	12.5
Psoriasis	7	3.0
Atopic dermatitis	9	3.9
Allergic/irritant dermatitis	22	9.5
Therapy at baseline		
Chemotherapy	100	43.1
Radiotherapy	46	19.8
Other	46	19.8
Complications		
Dermatitis	145	62.5
Pruritis/xerosis	140	60.3
Infection	25	10.8
Ulceration	82	35.3
Type of stoma		
Colostomy	95	41.3
Ileostomy	103	44.8
Urostomy	32	13.9

**Table 2 healthcare-11-01823-t002:** Variables associated with dermatitis.

		95% CI		
Variable	OR	LB	UB	*p* Value
BMI = Normal	0.488	0.229	1.042	0.064
Caregiver = Completely dependent	2.193	0.961	5.003	0.062
Psoriasis	33.324	2.175	510.555	0.012
Physical activity = Never or almost never	2.088	0.955	4.568	0.065
Diet = Low residue diet	0.222	0.084	0.587	0.002
Chemotherapy	2.237	1.084	4.613	0.029
Remover (accessory)	0.528	0.258	1.081	0.081
Powder (accessory)	2.170	0.924	5.096	0.075
Urostomy	0.251	0.093	0.680	0.007
Abdominal stoma profile = Regular	0.314	0.140	0.702	0.005
Type of device = One piece	3.763	1.536	9.219	0.004
Hydrocolloid plaque	0.108	0.041	0.282	<0.001
Gauze for cleansing	0.109	0.016	0.727	0.021
Low pH detergent	0.013	0.002	0.091	<0.001

OR, odds ratio; LB, lower boundary; UB, upper boundary.

**Table 3 healthcare-11-01823-t003:** Variables associated with pruritis/xerosis.

		95% CI		
Variable	OR	LB	UB	*p* Value
Male sex	2.954	1.584	5.507	0.001
BMI = Overweight or obese	2.341	1.195	4.583	0.013
Sedentary lifestyle	0.240	0.073	0.792	0.019
Ileostomy	1.798	0.956	3.382	0.069
Deep convex plaque	5.119	1.105	23.729	0.033
Hydrocolloid plaque + dressing in TNT	2.050	1.060	3.963	0.033
Age > 75 years	0.383	0.184	0.797	0.010

OR, odds ratio; LB, lower boundary; UB, upper boundary.

**Table 4 healthcare-11-01823-t004:** Variables associated with infections.

		95% CI		
Variable	OR	LB	UB	*p* Value
BMI = Class 2 obesity	30.313	5.744	159.974	0.001
Atopic dermatitis	6.423	0.948	43.533	0.057
Other medical therapies	4.473	1.527	13.100	0.006
Hernia	11.818	2.257	61.883	0.003
One-piece device	0.174	0.036	0.840	0.029
Caregiver = Completely dependent	2.989	1.046	8.538	0.041
Frequent physical activity	2.126	1.084	4.170	0.028

OR, odds ratio; LB, lower boundary; UB, upper boundary.

**Table 5 healthcare-11-01823-t005:** Variables associated with ulcerations.

		95% CI		
Variable	OR	LB	UB	*p* Value
BMI = Underweight	3.907	1.419	10.758	0.008
Caregiver = Completely dependent	3.376	1.645	6.928	0.001
IBD	4.161	1.307	13.248	0.016
Allergic/irritant dermatitis	2.345	0.842	6.531	0.077
Ileostomy	1.860	0.906	3.819	0.091
Radiotherapy	2.394	1.055	5.436	0.037
Regular abdominal stoma profile	0.391	0.190	0.801	0.009
Protective film (accessory)	0.421	0.207	0.854	0.018
Everted stoma	0.199	0.058	0.684	0.011
Normal stoma	0.252	0.098	0.649	0.009

OR, odds ratio; LB, lower boundary; UB, upper boundary.

## Data Availability

The data used in this study are available upon reasonable request to the corresponding author, subject to any ethical and legal constraints.

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
