# Peer review of "Peristomal Skin Complications: Detailed Analysis of a Web-Based Survey and Predictive Risk Factors"

_healthcare, 2023, doi:10.3390/healthcare11131823_

Round 1

Reviewer 1 Report

The authors present a descriptive analysis of PSC using a web survey and try to find descriptive risk factors.

1. The authors do not use a method to calculate sample sizes, as they acknowledge, which severely limits the reach of the study. Also, do the authors know the contribution from each nurse to the total n of patients? It is important that the authors describe how many patients were obtained by each nurse since this could be a severe bias of the study, as well. 

Also, is the lack of 4 months of follow-up the only exclusion criteria?

2. The authors use a stepwise regression model to determine the most important variables that could be associated with the outcomes observed. However, it has been described that stepwise regression might incur in some bias, during selection of variables by not taking into account the theoretical/practical relation of those variables and the outcome. In general, this model although sometimes useful, could provide variables that have statistical merit, but not necessarily a biological one. I know the authors mention the usage of only plausible risk factors, but do they consider that they might be excluding also relevant risk factors due to the stepwise selection? 

I ask the authors to describe the criteria for choosing the "plausible" risk factors and how they deal with the fact that some of the relevant variables might be excluded from this model just based on p-values but not a combination of biological value and p-values.

3. Did the authors explore other options for identifying risk factors? Do the authors believe that some of the risk associations they find are just an artifact of the model, or are they truly risk or protective factors? e.g. the authors mention that "we did not find ileostomy to be a risk factor for dermatitis but rather found urostomy to be protective in our cohort." Does this finding make sense from a biological point of view or is it just making sense because it fits the model used?

I ask the authors to describe if the relations they find are biologically relevant, not only mention as opposite from previous findings and if they believe they are part of the bias of the model find alternatives to describe variables with biological and significant value.

4. The authors define complications in 4 large and broad groups. Do the authors have a reason to not further divide into different type of dermatitis, or divide into viral, bacterial and fungal infections? Although the authors mention they do not have access to clinical records, is it possible to further subdivide? I believe that the biological component of the complications is omitted when they are grouped together (e.g. viral vs fungal vs bacterial infections). Please explain the reasoning behind this.

Reviewer 2 Report

The authors present the results of a cross-sectional survey on the occurrence of four different outcomes and their relationship with a number of risk factors. The topic is of interest, as optimum stoma care is both important for the quality of life and the health of the affected, and may reduce costs for the health care system by avoiding those treatment-related factors which are modifiable (many risk factors will be more or less non-modifiable). Publication is recommended after the following issues have been addressed: 

Some major issues:
- in line 79, a nested design is described: 23 stoma nurses report on 263 (or, in terms of fully eligible patients, 232) stoma patients. If the authors employ a multifactorial analysis, that is, logistic regression analysis, which does not take this nesting into account (as e.g. a mixed model could), they should briefly mention why they think they can ignore the hierarchical structure of data.  
- The authors choose to use a dichotomous outcome (occurrence or not of one of the four respective outcomes) which is certainly a valid approach. However, there was considerable spread of the time under observation: from the minimally required 4 months to 30 months, with mean 9.17, sd 3.8). I would expect that the probability of an outcome occurring is related to the time of follow-up, i.e., the longer the follow-up, the more likely the occurrence. I am not necessaryly suggesting to use time-to-event analysis (Cox proportional hazards model) instead, but would suggest to include the time of follow-up, possibly categorized according to the tertiles or quartiles of the distribution, as additional explanatory variable - or otherwise explain why this has not been deemed necessary in terms of control of potential confounding (e.g. in case of no association with the outcomes or no association with explanatory factors - the former at least would be against my expectation). If, and only if, all explanatory variables should have been documented at baseline, confounding by lenght of follow-up would not be possible, and it would then be a mere matter of scientific interest to include length of follow-up, or not.  

Some minor issues:
- in the abstract, "class 2 obesity" is mentioned. I am not aware of a generally accepted definition? Hence would prefer to indicate a BMI-range instead, like 30 to 40 or whatever is appropriate
- line 102: concerning "baseline characteristics", the time until 2 weeks after stoma surgery is mentioned. The authors should briefly outline which characteristics were concerned (I guess variables described in line 106 following) - and in contrast to which other "non-baseline" variables (I guess the outcomes, but additionally any explanatory factors possibly at the time of occurrence of the outcome?).
- line 111: what is "abdominal profile" or "profile options"? Please explain.
- line 208 and elsewhere: what is a "deep convex plaque"? Please explain. Generally double-check that all such technical terms are explained upon first occurrence, taking into account that the readership of "Healthcare" is likely often non-specialist/non-gastroenterologist
- line 191 following: the presentation of results both as table - e.g. Table 2 and Figure 2 (left) for the outcome "dermatitis" or Table 3 and figure 2 right for "pruritus/xerosis" and so on - is redundant. If opting for a table, please include the CI in parentheses after the point estimate, which reads much nicer. The p-value can be omitted, as (i) any quantitative information on the strenght (and direction) of the risk factor can be taken from the OR estimate and (ii) a CI not including unity will be equivalent to a significant Wald test (for dichotomous explanatory variables, that is)
- The OR figures are quite nice, but I would suggest to add some more, and appropriately log-scaled vertical reference lines between 0.1 and 1, and 1 and 10, respectively. The presently one reference line right in the middle of the interval (1;10) is e.g. at 3.16, which seems a bit odd. Besides, the figures need to be checked, as they do not seem to correspond to the information in the table. For instance, the risk factor radiotherapy for ulcerations is indicated to have an OR of 2.39 (95% CI: 1.06-5.44) in table 5, while in Figure 3 right, the lower CL clearly extends below the reference of 1.0. How have these figures been produced? This needs amendment, and in case of doubt the figures should be omitted if they cannot be amended.

The quality of the English language is fine. Independend of this, technical terms need to be better explained (see comments above for examples).

Round 2

Reviewer 1 Report

I thank the authors for addressing most of the relevant comments. However, I still ask the authors to make some further clarifications before the final publication.

1. Please, make sure to comment in the manuscript the reason why you did not split the groups into different subgroups: "Due to the nature of logistic regression, a sufficient number of events is necessary for the correct estimation of parameters, thus making it necessary to group similar phenomena. It was thus not possible to obtain statistically relevant results to further differentiate the type of dermatitis"

2. Also, I believe the authors did not include the response to this question: "I ask the authors to describe the criteria for choosing the "plausible" risk factors and how they deal with the fact that some of the relevant variables might be excluded from this model just based on p-values but not a combination of biological value and p-values."

Please make sure to include this as part of your discussion and as a possible limitation of your studies. 

3. Finally, please make sure to highlight in your discussion the limitations of the stepwise regression model as part of the general limitations of the manuscript. 

In general, the manuscript can be accepted if the authors make these corrections. 

Author Response

Reviewer1: I thank the authors for addressing most of the relevant comments. However, I still ask the authors to make some further clarifications before the final publication.

  1. Please, make sure to comment in the manuscript the reason why you did not split the groups into different subgroups: "Due to the nature of logistic regression, a sufficient number of events is necessary for the correct estimation of parameters, thus making it necessary to group similar phenomena. It was thus not possible to obtain statistically relevant results to further differentiate the type of dermatitis”

We acknowledge that further investigation into specific types of dermatitis could provide valuable insights. However, given the limitations of the available data and the logistic regression requirements, it was impossible to achieve statistically meaningful results through subgroup analysis. We have discussed these limitations in the manuscript to provide transparency and clarity regarding our study design and analysis choices.

  1. Also, I believe the authors did not include the response to this question: "I ask the authors to describe the criteria for choosing the "plausible" risk factors and how they deal with the fact that some of the relevant variables might be excluded from this model just based on p-values but not a combination of biological value and p-values."

Please make sure to include this as part of your discussion and as a possible limitation of your studies. 

Dear reviewer, we clarified our approach in the methods as follows: “Independent variables were included in the models based on their p-values in relation to the outcome, and their selection was further supported by a literature search to verify the hypothetical clinical and biological association between the independent variables and outcomes. This combined approach represents a tradeoff between a data-driven approach and a theory-driven approach, allowing for a comprehensive selection of independent variables that considered both statistical significance and prior knowledge from the literature.”

  1. Finally, please make sure to highlight in your discussion the limitations of the stepwise regression model as part of the general limitations of the manuscript. 

We appreciate the reviewer’s comment and acknowledge the limitations of the stepwise regression model used in our study. The rationale behind adopting the stepwise technique was two-fold: first, to maintain a manageable number of independent variables in the logistic regression model, given the limited number of observed statistical units, and second, to address potential issues of multicollinearity among the independent variables.

The stepwise regression technique can achieve model parsimony by selecting only risk factors that demonstrate a statistically significant impact on the dependent variable. However, it is important to acknowledge that this approach may have a drawback, as it is possible that some factors that are indeed risk factors in the population may not be included in the final model. We acknowledged this possible drawback in the paragraph on limitations.

Reviewer 2 Report

Thank you for partially revising your paper (or at least rendering text color red, as in the introduction, which seems mostly unchanged - better to use "track changes" option of your word processor).

Concerning my previous major issues: (i) The authors basically claim that the nested design does not affect "individuality" of data. This would require that all involved nurses have been trained, or follow the same standards when administering the questionnaire of baseline/explanatory variables on their respective patients. This must be mentioned in the "Methods". Concerning the outcome, I wonder whether nurses were actively involved in the stoma care of the patients who were entered in the study database, or just observers and counsellors of the patients? In the former case I would find it bold to assume no impact of which nurse cared for the patient, and entered his/her data, and would request to mention the impossibility to address the impact of the nurse "as risk factor" as limitation in the discussion (as the authors seem to be unwilling to in any way expand their analysis). (ii) I am not sure I understand the authors' explanation that no information on the time of follow-up exists (which could be correlated with outcome occurrence) as they themselves offer a description of this time. Anyway, consideration of time is not a must, as all explanatory variables have been recorded at baseline (as now made clear) and hence confounding by time is impossible.

The minor issues have been addressed adequately, except that the abbreviations "OR", "LB", and "UB" in the tables with logistic regression results need to be explained in each of the table legends. BTW, which confidence level did the autors use? Rhetorical question, as it can be read 100 times in the text - but it must also be mentioned in the table with its legend as this needs to be intellegible "stand alone" (basic scientific writing).

none

Author Response

Reviewer2: Thank you for partially revising your paper (or at least rendering text color red, as in the introduction, which seems mostly unchanged - better to use "track changes" option of your word processor).

Concerning my previous major issues: 

(i) The authors basically claim that the nested design does not affect "individuality" of data. This would require that all involved nurses have been trained, or follow the same standards when administering the questionnaire of baseline/explanatory variables on their respective patients. This must be mentioned in the "Methods". Concerning the outcome, I wonder whether nurses were actively involved in the stoma care of the patients who were entered in the study database, or just observers and counsellors of the patients? In the former case I would find it bold to assume no impact of which nurse cared for the patient, and entered his/her data, and would request to mention the impossibility to address the impact of the nurse "as risk factor" as limitation in the discussion (as the authors seem to be unwilling to in any way expand their analysis).

We appreciate the reviewer’s valuable comment and concern regarding the individuality of the data and the potential impact of the nurses involved in the study. We agree that addressing these points in the Methods section and discussing them as limitations in the manuscript is essential for clarity and transparency.

In the methods, we added, “In addition, all participating nurses administering the survey for gathering data about their patients had a standardized base of competence and had attended the same certifications in stoma care. Therefore, a nested structure in considering the data was considered unlikely, and the data were considered at an individual level without the need for nested models.”

In the limitations, we added this statement: “While the likelihood of a hypothetical nurse-outcome relationship was considered highly limited when the study was designed, it is necessary to recognize that the potential impact of the individual nurse who cared for the patient and entered the data cannot be completely disregarded. However, considering the standardized base of competence of the involved nurses as a framework, the collected data were considered at an individual level, which was deemed appropriate for the nature of this study. This approach allows for future designs to incorporate more robust methodologies. Although we acknowledge this limitation, it is important to note that as an observational study, our analysis focused on the relationship between the identified risk factors and the outcomes of interest, without explicitly examining the impact of individual nurses as a separate risk factor.”

(ii) I am not sure I understand the authors' explanation that no information on the time of follow-up exists (which could be correlated with outcome occurrence) as they themselves offer a description of this time. Anyway, consideration of time is not a must, as all explanatory variables have been recorded at baseline (as now made clear) and hence confounding by time is impossible.

We understand your point about the potential correlation between the time of follow-up and the occurrence of the outcome. While we did provide a description of the time under observation, we meant to convey that we did not have specific information on the time at which each outcome occurred for individual patients. As a result, we were unable to account for the exact timing of outcome occurrence during the follow-up period. In our study, all the explanatory variables were indeed recorded at baseline, providing a snapshot of the patient's characteristics at that specific time point. Considering the time of follow-up as an additional explanatory variable, categorized according to tertiles or quartiles was not deemed necessary to control for potential confounders. This decision was based on the understanding that confounding by the length of follow-up would not be possible since all explanatory variables were documented at baseline.

Including the length of follow-up as an explanatory variable is interesting but go beyond the scope of our study, which was aimed at creating a risk profile based on available factors at baseline. Moreover, incorporating time as an additional variable would not be consistent with the concept of a risk profile, which represents an estimation made at time 0 with the factors available at time 0.

We appreciate your suggestion, but given the study design and the focus on baseline characteristics, the inclusion of length of follow-up was not considered necessary for our analysis.

As per your comment, we also clarified these aspects in the current version of the study aims, and we extensively enriched the limitations paragraph and the elements to interpret the current results in the most prudent and adequate way.

Round 3

Reviewer 1 Report

I thank the authors for their efforts and improvement of the manuscript.

The manuscript is suitable for publication with a minor addition. I please ask the authors to include the p-values in their tables to make it easier to the readers, instead of reading al the values only from the text.

Author Response

Dear Reviewer, 

We have added the p values to the tables as requested. Many thanks.

Reviewer 2 Report

Thank you for your latest revision and explanations in the manuscript - fine now.

English language is largely fine.

Author Response

Thanks to the Reviewer.